Fatty acid profiles of highly migratory resources from the Southeastern Pacific Ocean, Chile: a potential tool for biochemical and nutritional traceability

Guzmán-Rivas Fabián 1
Quispe-Machaca Marco 1
Lazo Jorge 2
Ortega Juan Carlos 3
Mora Sergio 3
Barría Martínez Patricio 4
Urzúa Ángel aurzua@ucsc.cl 2 5
1 Programa de Doctorado en Ciencias con mención en Biodiversidad y Biorecursos, Facultad de Ciencias, Universidad Católica de la Santísima Concepción , Concepción , Biobío , Chile
2 Centro de Investigación en Biodiversidad y Ambientes Sustentables (CIBAS), Universidad Católica de la Santísima Concepción , Concepción , Biobío , Chile
3 Instituto de Fomento Pesquero (IFOP) , Talcahuano , Biobío , Chile
4 Instituto de Fomento Pesquero (IFOP) , Valparaíso , Valparaíso , Chile
5 Departamento de Ecología, Facultad de Ciencias, Universidad Católica de la Santísima Concepción , Concepción , Biobío , Chile
Esteban María Ángeles
Electronic publication date: 2025 Mar 20
Publication date: 2025
Volume: 13
Electronic Location ID: e19101
Received 2024 Jan 16; Accepted 2025 Feb 12
Copyright: ©2025 Guzmán-Rivas et al.
Copyright year: 2025
Copyright holder: Guzmán-Rivas et al.
License: This is an open access article distributed under the terms of the Creative Commons Attribution License, which permits unrestricted use, distribution, reproduction and adaptation in any medium and for any purpose provided that it is properly attributed. For attribution, the original author(s), title, publication source (PeerJ) and either DOI or URL of the article must be cited.
License URL: https://creativecommons.org/licenses/by/4.0/

Keywords: Fishery, Migratory Species, Traceability, Lipids, Fatty acids

Funding: Fisheries Project of Highly Migratory Resources: Biological-Fishing Aspects 36547-102 Ciencia abierta en la UCSC INCA 210005 Fondo de Mantención de Equipamiento para la Investigación DI-FMEI 02/2023 This work was supported by the “Fisheries Project of Highly Migratory Resources: Biological-Fishing Aspects” (IFOP No. 36547-102). The work was further supported through the UCSC project: Ciencia abierta en la UCSC (grant INCA 210005) and Fondo de Mantención de Equipamiento para la Investigación (grant DI-FMEI 02/2023). The funders had no role in study design, data collection and analysis, decision to publish, or preparation of the manuscript.

==============================
The traceability of fish species and their resulting food products is essential to maintain the global supply of these goods, allowing us to distinguish and reconstruct the origin and history of their production chain. One way to trace food is through biochemical determinations, which aid in identifying their geographical origin quickly. This study analyzed the fatty acid (FA) profiles of highly migratory fishery resource species (HMRS) from the Southeastern Pacific Ocean (SEPO), and their use as potential tools to determine the geographic origin and nutritional condition of these marine resources. The fatty acids (FAs) presented in fillet or muscle tissue of 18 HMRS were measured as FA methyl esters by gas chromatography. Our results reveal that the swordfish Xiphias gladius presented the greatest variety of FAs, strongly characterized by the presence of saturated, monounsaturated, and polyunsaturated FAs. A similar trend of high diversity in all classes of FAs was observed in tuna species (i.e., Thunnus alalunga; T. albacares; T. obesus), oilfish (Ruvettus pretiosus) and escolar fish (Lepidocybium flavobrunneum). In turn, Lampris guttatus, Makaira indica, and Tetrapturus audax presented an intermediate variety of FAs and the highest amount of saturated and monounsaturated FAs of the evaluated species. Finally, Luvarus imperialis, Coryphaena hippurus and the sharks (Lamna nasus; Alopias vulpinus; Prionace glauca; Isurus oxyrinchus; Sphyrna zygaena) presented a low diversity of FAs, with only saturated FAs strongly predominating. Regarding the total concentration of FAs, the highest average values were recorded in X. gladius, L. flavobrunneum and R. pretiosus. The present study revealed notable differences in the FA compositions of the muscle of diverse HMRS from the SEPO off the coast of Chile, with the swordfish showing the healthiest FAs (i.e., mono and polyunsaturated) for human consumption. The data on FAs collected for HMRS could be used as a reference to characterize the FA profiles of other fisheries in the SEPO (e.g., coastal pelagic fishes). In an ecosystem approach, our findings help us to understand how essential nutrients (i.e., FA biomolecules) are transferred through the marine food web in the SEPO, revealing the diet type and/or feeding habits of HMRS considered as top predators. Furthermore, identifying the FA profiles of fishery resources at a spatial level provides crucial information for their management and conservation, particularly in those resources that are overexploited and also have a critical nutritional importance for human consumption.

Introduction

Traceability is an essential concept defined as “the possibility of finding and following the trace, through all the stages of production, transformation and distribution, of a food, a feed, an animal destined for production of food or a substance intended to be incorporated into food or feed or likely to be” (Badia-Melis, Mishra & Ruiz-García, 2015; Salampasis, Tektonidis & Kalogianni, 2012). In this way, we will consider traceability as a tool that allows us to distinguish and reconstruct the species, its origin, nutritional value, and the history of the production chain of a food product, recognizing all its phases, including the place of capture, harvest, production, processing, storage, and distribution, among others (Badia-Melis, Mishra & Ruiz-García, 2015; Bosona & Gebresenbet, 2013). In the current context of the Sustainable Development Goals (food security and life below water), “the food and/or nutritional traceability” of fishery resources by means of biochemical determinations (e.g., fatty acid profiles) can help in identifying the geographical origin quickly, while also revealing the nutritional quality of these marine resources (FAO, 2024).

Currently, the identification of highly migratory species from the Southeastern Pacific Ocean (SEPO) in Chile (Xiphias gladius, Prionace glauca, Isurus oxyrinchus, Lamna nasus, and Coryphaena hippurus) is carried out through species identification guides prepared with taxonomic criteria specific to each analyzed species (Parot, 2020), using information on the morphology of adult specimens and conspicuous characteristics, whether these are meristic or morphometric. Problems in species identification arise when a species must be classified by visible body parts alone (e.g., trunk or fins of a shark). For example, in the case of tuna, it is difficult to carry out a taxonomic recognition based on external morphology; in fact, it is feasible to identify species based on the morphology of the liver and the composition of its fats or through molecular analysis (Pedrosa-Gerasmio, Babaran & Santos, 2012; Roungchun, Tabb & Hellberg, 2022).

In addition, the problem of identifying highly migratory species occurs more frequently during the unloading process (i.e., when these marine resources arrive at the fishing port for sale), and when only the trunk is available (in gutted specimens without heads, tails or fins) (Clark, 2015; Miller, Jessel & Mariani, 2012). In this case, other morphological identification characters must be used, such as the presence of keels, notches, coloration, denticles on the skin, and, in some cases, the distribution of fats in cross-sections of the fillet or muscle tissue (Hernández et al., 2010; Mello, De Carvalho & Brito, 2013; Stefanni et al., 2021).

Seafood can be traced through the use of different tools (Ricardo et al., 2015). The identification and quantification of fatty acids is an effective tool to trace marine products (Ricardo et al., 2017; Rohman et al., 2023). Fatty acids are key molecules in organisms because they are part of the cell membrane, they participate in different physiological processes and are an important energy source (Dalsgaard et al., 2003; Samovski, Jacome-Sosa & Abumrad, 2023; Tocher & Glencross, 2015). These molecules can be affected by the species, their diet or habitat conditions and can even vary among different species, different stages of development or different seasons of the year (Sargent et al., 1999). Despite the variability of fatty acids, these can trace the geographic origin of a product because fatty acids are closely related to organisms lower on the food chain, such as phytoplankton or zooplankton (Dalsgaard et al., 2003; Fonseca et al., 2022). Within a marine trophodynamics context, it has been described that long-chain polyunsaturated fatty acids (LC-PUFAs) are predominantly available in high concentrations at primary or secondary levels of the marine food web. Top predators, such as highly migratory fish, do not have a strong capacity to biosynthesize LC-PUFAs, therefore, these essential biomolecules are obtained exclusively through diet and/or prey consumed at lower levels of the food web, and are consequently stored conservatively in predator tissues (Dalsgaard et al., 2003; Tan, Zhang & Zheng, 2022). This further confirms the importance of applying new and innovative methods for the biochemical traceability of marine species and fishery products of highly migratory resources in the Southeastern Pacific Ocean off the coast of Chile. In particular, biochemical determinations allow us to identify the species and its origin quickly and efficiently, from its origin in the marine food web until immediately after the unloading process.

It has been found that in highly productive marine ecosystems, such as the Humboldt Current System, highly migratory species considered as top predators (i.e., sharks, tuna, marlins, swordfish), demonstrate a link between their diet (prey consumption) and the fatty acid profiles of their organs and/or tissues (Barría et al., 2020; Lazo-Andrade et al., 2023; Quispe-Machaca et al., 2021; Segura-Cobeña et al., 2021; Hu et al., 2022). In particular, the lipids and/or fats of the prey are directly transferred and stored conservatively in the predators’ organs (Beckmann et al., 2013; Xu, Pethybridge & Li, 2022; Meyer et al., 2019). Subsequently, as required, these lipids are mobilized and used for various key physiological functions within their ontogeny (e.g., growth, reproduction, homeostasis) (Leigh, Papastamatiou & German, 2017; Alderete-Macal, Caraveo-Patiño & Hoyos-Padilla, 2020; Meyer et al., 2017; Meyer et al., 2021). Within highly migratory predator species, different eating habits and hunting behaviors of their prey have also been described. For example, in the case of shark species (e.g., P. glauca, I. oxyrinchus), tuna (e.g., Thunnus albacares, T. alalunga), and C. hippurus a voracious and generalist habit has been reported (Méndez, 2021). In turn, in marlin species (e.g., Makaira indica, Tetrapturus audax) and X. gladius the consumption of cephalopod species has been described as the main prey (Lazo-Andrade et al., 2021; Lazo-Andrade et al., 2023; Zhang et al., 2023). In particular, X. gladius has demonstrated a specialist hunting habit for the jumbo squid (Ibáñez, González & Cubillos, 2004; Lazo-Andrade et al., 2021); swordfish have been reported to first stab and then cut their prey with the sword of their mouth structure (Berkovitz & Shellis, 2017; Preti et al., 2023).

Most of these bony and cartilaginous fish species present a trophic and/or reproductive migratory pattern (for a conceptual model see: Lazo-Andrade et al., 2021; Lazo-Andrade et al., 2023; Barría et al., 2020) from equatorial water zones to cold temperate zones of the SEPO. These fishes aim to consume high-energy value prey necessary to sustain the metabolic expenditure involved in traveling long distances, as well as storing sufficient energy for their subsequent reproduction (Guzmán-Rivas et al., 2023; Segura-Cobeña et al., 2021; Meekan et al., 2022). These bioenergetic-adaptive attributes are characterized by the consumption of prey with a specific profile and/or fatty acid footprint that are also rich in lipids, and the posterior conservative storage of the energy from these prey in the organs. Fatty acid profiles can thus be used as a biochemical and nutritional traceability tool for these species that support important fisheries in the Humboldt System (Barría et al., 2020; Lazo-Andrade et al., 2023; Santos et al., 2023).

The study model species (Table 1, Fig. 1) are considered top predators that consume prey with high amounts of fat and/or energy. Consequently, the fatty acid profiles in their tissues reflect the food consumed, which is transferred and stored conservatively in their muscle. Therefore, we hypothesized that the fatty acid profiles may vary, not only depending on the prey consumed through a characteristic feeding habit and/or hunting behavior, but also due to their phylogeny and/or taxonomic group of evolutionary origin. It is thus expected that the model species studied present interspecific variations in the fatty acid profiles of their muscle. This tool can be useful not only to identify the biochemical traceability of fish species at the tissue level, but also their nutritional condition and fillet quality. Therefore, in the present study, we analyzed the fatty acid composition in highly migratory species obtained from Humboldt marine ecosystem off the coast of Chile.

Table 1 Experimental design and name of each species analyzed.

Species with n < 3 were not considered in the statistical analysis.

Scientific name	Common name	n	
Alopias vulpinus	Common thresher	4	
Coryphaena hippurus	Mahi-mahi	8	
Gasterochisma melampus	Butterfly kingfish	2	
Istiompax indica	Black marlin	1	
Isurus oxyrinchus	Shortfin mako shark	8	
Kajikia audax	Striped marlin	1	
Katsuwonus pelamis	Skipjack tuna	2	
Lamna nasus	Porbeagle shark	8	
Lampris guttatus	Sunfish	8	
Lepidocybium flavobrunneum	Escolar fish	9	
Luvarus imperialis	Louvar	4	
Prionace glauca	Blue shark	8	
Ruvettus pretiosus	Oilfish	7	
Sphyrna zygaena	Hammerhead shark	1	
Thunnus alalunga	Albacore	5	
Thunnus albacares	Yellowfin Tuna	7	
Thunnus obesus	Bigeye tuna	8	
Xiphias gladius	Swordfish	27	

Figure 1 Illustrations of different highly migratory species caught off the coast of Chile.

Materials and Methods

Ethical declaration

This study was guided in accordance with the Act on Welfare and Management of Marine Animals, and they comply with the current Chilean animal care and manipulation legislation of the fishery resources (SUBPESCA). Consequently, to prevent the pain of the fishes during their collection and handling, they were put to sleep with a cold shock (Law 20.380, Ministry of Health and Ethics Committee, Chile) (Robb & Kestin, 2002).

Sample collection and analyzed species

Scientific observers from the Fisheries Development Institute-IFOP, as part of the Highly Migratory Fisheries Resources Project (Biological-Fisheries Aspects (IFOP, MINECON)), collected tissue samples (muscle) of highly migratory species captured on vessels of the trawler and industrial longline fishing fleets targeting swordfish; some species were included in the bycatch category (Barría et al., 2020; Guzmán-Rivas et al., 2023) (Table 1).

All specimens were identified, measured, weighed, and sexed on board the boats.

A total of 18 HMRS from the SEPO were analyzed (Fig. 2), of which three corresponded to tuna species (i.e., albacore, yellowfin, bigeye), five to shark species (i.e., porbeagle, common thresher, blue, shortfin mako, hammerhead), eight to other species of large fish (i.e., mahi-mahi, swordfish, louvar, escolar, skipjack, oil, butterfly kingfish, sun) and two to species of billfish (i.e., black, striped). For more details, including scientific names, common names and number of samples analyzed per species, see (Table 1, Fig. 1).

Figure 2 Sampling point of different highly migratory species caught off the coast of Chile.

Muscle samples were taken between the first and second dorsal fins at a depth of approximately 20 mm from the skin to the underlying connective tissue or muscle tissue for cartilaginous fish; for bony fish these samples were extracted from the trunk (for details on procedures, see: Barría et al., 2020; Lazo-Andrade et al., 2021; Lazo-Andrade, Barría & Urzúa, 2024). After obtaining the samples, they were immediately placed on dry ice to be transported and stored at −80 °C in a freezer until the subsequent biochemical traceability analysis (measured as their fatty acid compositions) in the Hydrobiological Resources Laboratory of the Universidad Católica de la Santísima Concepción, Chile. Here, the muscle samples from highly migratory fishery resource species were vacuum- and cold-dried at −80 °C in a freeze-dryer (Operon, FDU-7012). Then, they were sonicated (MRC, AC-120H) and stored in 15 mL centrifuge tubes for the immediate analysis of their fatty acid compositions.

Fatty acid compositions

First, the lipid content was extracted from 20 mg of the dry weight (DW) using the gravimetric method developed by Cequier-Sánchez et al. (2008). This method used dichloromethane:methanol (2:1) as an organic solvent. Then, the fatty acid compositions were determined following the method used by Malzahn et al. (2007), where fatty acid methyl esters (FAMEs) were measured from the total lipids extracted. These lipids were esterified by incubation with methanolic sulfuric acid in a Thermo Shaker (Model: DBSO-001) at 70 °C for 1 h. Then, the fatty acids were obtained by washing with n-Hexane. This washing was performed three times using six mL, three mL, and three mL of the respective solvent. At the end of each wash, each mixture was mixed using a vortex, and the upper phase was added to a new previously primed amber bottle. Finally, the fatty acids were concentrated using a sample concentrator (MD 200) supplemented with nitrogen gas. The FAMEs were measured using a gas chromatograph (Agilent, model 7890A) at a set temperature with a DB-225 column (J and W Scientific, 30 m long, 0.25 mm interior diameter, and 0.25 µm thick film). FAMEs were quantified using the C23:0 fatty acid as an internal standard according to Malzahn et al. (2007), with Agilent chromatography software (ChemStation). FAMEs were then identified by comparison with known standards of fatty acids of marine origin (using certified material, Supelco 37, mixture FAME 47,885 –U).

Statistical analysis

Statistical tests were performed using the PRIMER-E (PRIMER6 v.6.1.16 and PERMANOVA + v.1.0.6) program. The advanced statistical analyses (i.e., nonparametric multivariate tests) described by Zuur, Leno & Smith (2007), Clarke & Gorley, 2015; with a confidence level of 95% (p < 0.05) were also applied. To identify the species through their species/specific fatty acid profiles, a Bray-Curtis similarity and similarity matrix were carried out with their respective multivariate analyses: (i) multidimensional scaling (MDS) and (ii) principal coordinates (PCoA). Prior to analysis, the data set was transformed (i.e., square root) and a resemblance test (i.e., Bray-Curtis) was carried out. In addition, a similarity analysis (ANOSIM) was performed to determine similar species and/or groups with a determination coefficient close to zero (R = 0). For very different species and/or groups, a determination coefficient close to one (R = 1) was utilized. Subsequently, a similarity percentage analysis (SIMPER) was carried out to determine the percentage of contribution of the different types of fatty acids in the different species analyzed. Finally, a multivariate permutation analysis (PERMANOVA) was used to compare the fatty acid profiles of the diverse species. Species with n < 3 were not considered in the statistical analyses (for number and name of species analyzed, see Table 1).

Results

Fatty acid compositions

The fatty acid compositions presented highly significant differences among the analyzed species (PERMANOVA, Pseudo-F= 18.875; P < 0.001). That is, a distinctive and characteristic fatty acid composition was identified for each study species; these biochemical/physiological traits are reflected in the amount (i.e., concentration) and variety (i.e., different types) of fatty acids observed in the muscle (tissue samples of 20 mg of DW) of each species.

In terms of variety and composition of fatty acids, X. gladius consistently presented the greatest variety of fatty acids, strongly characterized by the presence of all classes of fatty acids (saturated, monounsaturated, polyunsaturated-n3, and polyunsaturated-n6). A similar trend of high variety in all classes of fatty acids was observed in tuna species (T. alalunga, T. albacares, T. obesus), Ruvettus pretiosus and Lepidocybium flavobrunneum. In turn, Lampris guttatus and marlins (M. indica, T. audax) presented an intermediate variety of fatty acids, though a greater presence of saturated and monounsaturated fatty acids was noteworthy in these species. Finally, Luvarus imperialis, C. hippurus and the sharks (L. nasus, Alopias vulpinus, P. glauca, I. oxyrinchus, Sphyrna zygaena) presented a low variety of fatty acids, where only saturated fatty acids strongly predominated. Furthermore, muscle samples of some species in which only a few specimens were analyzed (<3 individuals), indicated that Gasterochisma melampus presented a medium variety of fatty acids (12 different fatty acids), while Katsuwonus pelamis presented the lowest variety of fatty acids (three fatty acids). For more details regarding the different types of fatty acids detected in the diverse species see Table 2.

Table 2 Fatty acid profile of muscle tissue of N = 18 highly migratory species off the coast of Chile.

The data set are represented as average values (expressed in 20 mg dry weight) and standard deviation (AV ± SD), while the bold values indicate the total sum and standard deviation for each fatty acids class.

Fatty acids	Alopias vulpinus	Coryphaena hippurus	Gasterochisma melampus x	Istiompax indica x	Isurus oxyrinchus	Kajikia audax x		
C11:0	n.d.	n.d.	n.d.	n.d.	n.d.	n.d.	
C13:0	n.d.	n.d.	0.01 ± 0.001	n.d.	n.d.	n.d.	
C14:0	n.d.	n.d.	0.01 ± 0.002	0.04*	n.d.	0.1*	
C15:0	n.d.	n.d.	n.d.	0.01*	n.d.	0.02*	
C16:0	0.02 ± 0.003	0.02 ± 0.002	0.04 ± 0.02	0.21*	0.03 ± 0.01	0.47*	
C17:0	n.d.	n.d.	n.d.	0.01*	n.d.	0.02*	
C18:0	0.01 ± 0.001	0.01 ± 0.001	0.02 ± 0.01	0.1*	0.02 ± 0.003	0.25*	
C20:0	n.d.	n.d.	n.d.	n.d.	n.d.	0.01*	
C22:0	n.d.	n.d.	n.d.	n.d.	n.d.	n.d.	
C23:0	n.d.	n.d.	n.d.	n.d.	n.d.	n.d.	
Σ SFA	0.03 ± 0.003	0.03 ± 0.003	0.08 ± 0.02	0.37 ± 0.08	0.05 ± 0.01	0.87 ± 0.18	
C14:1	n.d.	n.d.	n.d.	n.d.	n.d.	n.d.	
C16:1	0.01 ± 0.001	n.d.	0.03*	0.03*	0.01 ± 0.002	0.09*	
C17:1	n.d.	n.d.	0.02*	0.01*	n.d.	0.02*	
C18:1n9	n.d.	0.02 ± 0.003	0.69 ± 0.44	0.19*	0.02 ± 0.003	0.52*	
C20:1	n.d.	n.d.	0.1 ± 0.003	0.01*	n.d.	0.02*	
C22:1n9	n.d.	n.d.	0.02 ± 0.003	n.d.	n.d.	n.d.	
C24:1	n.d.	n.d.	n.d.	0.01*	n.d.	0.01*	
Σ MUFA	0.01 ± 0.001	0.02 ± 0.003	0.86 ± 0.36	0.25 ± 0.08	0.03 ± 0.01	0.66 ± 0.22	
C18:3n3	n.d.	n.d.	n.d.	n.d.	n.d.	0.01*	
C20:5n3 (EPA)	n.d.	n.d.	0.04 ± 0.004	0.01*	n.d.	0.03*	
C22:6n3 (DHA)	n.d.	0.01 ± 0.003	0.07 ± 0.001	0.04*	0.02 ± 0.01	0.07*	
Σ PUFA-n3	n.d.	0.01 ± 0.003	0.11 ± 0.06	0.05 ± 0.02	0.02 ± 0.01	0.11 ± 0.03	
C18:2n6c	n.d.	n.d.	n.d.	n.d.	n.d.	n.d.	
C18:3n6	n.d.	n.d.	0.02 ± 0.003	0.01*	n.d.	0.02*	
C20:2n6	n.d.	n.d.	n.d.	n.d.	n.d.	n.d.	
C20:3n6	n.d.	n.d.	0.03	n.d.	n.d.	0.01*	
Σ PUFA-n6	n.d.	n.d.	0.05 ± 0.004	0.01*	n.d.	0.03 ± 0.01	
Σ PUFA	n.d.	0.01 ± 0.003	0.16 ± 0.04	0.06 ± 0.02	0.02 ± 0.01	0.14 ± 0.02	
Total FA	0.4 ± 0.005	0.06 ± 0.004	1.1 ± 0.21	0.68 ± 0.06	0.1 ± 0.01	1.67 ± 0.15	
Katsuwonus pelamisx	Lamna nasus	Lampris guttatus	Lepidocybium flavobrunneum	Luvarus imperialis	Prionace glauca	
n.d.	n.d.	n.d.	0.02 ± 0.001	n.d.	n.d.	
n.d.	n.d.	n.d.	0.03 ± 0.01	n.d.	n.d.	
n.d.	n.d.	0.03 ± 0.02	0.03 ± 0.01	n.d.	n.d.	
n.d.	n.d.	0.01 ± 0.001	n.d.	n.d.	n.d.	
0.03 ± 0.002	0.04 ± 0.01	0.09 ± 0.01	0.09 ± 0.06	0.02 ± 0.002	0.03 ± 0.01	
n.d.	n.d.	0.01 ± 0.002	n.d.	n.d.	n.d.	
0.01 ± 0.002	0.02 ± 0.003	0.04 ± 0.03	0.05 ± 0.01	0.01 ± 0.001	0.02 ± 0.01	
n.d.	n.d.	n.d.	n.d.	n.d.	n.d.	
n.d.	n.d.	n.d.	n.d.	n.d.	n.d.	
n.d.	n.d.	n.d.	0.06 ± 0.01	n.d.	n.d.	
0.04 ± 0.01	0.06 ± 0.01	0.18 ± 0.06	0.28 ± 0.05	0.03 ± 0.003	0.05 ± 0.01	
n.d.	n.d.	0.01*	n.d.	n.d.	n.d.	
n.d.	n.d.	0.04 ± 0.04	0.14 ± 0.7	0.01 ± 0.001	0.01 ± 0.01	
n.d.	n.d.	0.01 ± 0.001	0.09 ± 0.08	n.d.	n.d.	
0.03*	0.02 ± 0.002	0.18 ± 0.02	2.04 ± 0.88	0.02 ± 0.001	0.03 ± 0.01	
n.d.	n.d.	0.05 ± 0.03	0.21 ± 0.11	n.d.	n.d.	
n.d.	n.d.	n.d.	0.05 ± 0.01	n.d.	n.d.	
n.d.	n.d.	n.d.	0.06 ± 0.01	n.d.	n.d.	
0.03*	0.02 ± 0.002	0.29 ± 0.14	2.59 ± 0.88	0.03 ± 0.004	0.04 ± 0.01	
n.d.	n.d.	n.d.	0.05 ± 0.02	n.d.	n.d.	
n.d.	n.d.	0.02 ± 0.01	0.08 ± 0.04	n.d.	0.01 ± 0.001	
n.d.	n.d.	0.02 ± 0.01	0.2 ± 0.12	n.d.	0.02 ± 0.01	
n.d.	n.d.	0.04 ± 0.01	0.33 ± 0.11	n.d.	0.03 ± 0.01	
n.d.	n.d.	n.d.	n.d.	n.d.	n.d.	
n.d.	n.d.	0.03 ± 0.02	0.06 ± 0.03	n.d.	n.d.	
n.d.	n.d.	n.d.	0.03 ± 0.01	n.d.	n.d.	
n.d.	n.d.	n.d.	0.09 ± 0.04	n.d.	0.01*	
n.d.	n.d.	0.03 ± 0.02	0.18 ± 0.04	n.d.	0.01*	
n.d.	n.d.	0.07 ± 0.01	0.51 ± 0.09	n.d.	0.04 ± 0.01	
0.07 ± 0.01	0.08 ± 0.01	0.54 ± 0.09	3.38 ± 0.57	0.06 ± 0.004	0.13 ± 0.01	
Ruvettus pretiosus	Sphyrn zygaena x	Thunnus alalunga	Thunnus albacares	Thunnus obesus	Xiphias gladius	
0.01 ± 0.003	n.d.	n.d.	n.d.	n.d.	n.d.	
0.03 ± 0.01	n.d.	n.d.	0.01*	n.d.	n.d.	
0.02 ± 0.01	n.d.	0.05 ± 0.03	0.03 ± 0.01	0.03 ± 0.02	0.11 ± 0.07	
n.d.	n.d.	0.02 ± 0.004	0.01 ± 0.01	0.01 ± 0.01	0.03 ± 0.02	
0.08 ± 0.05	0.02*	0.3 ± 0.24	0.13 ± 0.09	0.11 ± 0.1	0.59 ± 0.4	
n.d.	n.d.	0.03 ± 0.01	0.02 ± 0.01	0.01 ± 0.01	0.03 ± 0.02	
0.05 ± 0.03	0.01*	0.09 ± 0.07	0.05 ± 0.04	0.05 ± 0.4	0.21 ± 0.14	
n.d.	n.d.	0.01*	0.02*	0.01*	0.05 ± 0.02	
n.d.	n.d.	n.d.	n.d.	n.d.	0.06 ± 0.03	
0.04 ± 0.03	n.d.	n.d.	n.d.	n.d.	0.05 ± 0.02	
0.23 ± 0.04	0.03 ± 0.003	0.05 ± 0.15	0.27 ± 0.1	0.22 ± 0.07	1.13 ± 0.26	
n.d.	n.d.	n.d.	n.d.	n.d.	n.d.	
0.05 ± 0.03	n.d.	0.06 ± 0.02	0.03 ± 0.02	0.03 ± 0.03	0.15 ± 0.1	
n.d.	n.d.	0.02 ± 0.01	n.d.	0.01 ± 0.01	0.04 ± 0.02	
1.36 ± 1.04	0.02*	0.23 ± 0.19	0.17 ± 0.1	0.16 ± 0.1	1.31 ± 0.84	
0.21 ± 0.16	n.d.	0.08 ± 0.02	0.04 ± 0.02	0.04 ± 0.02	0.19 ± 0.12	
0.06 ± 0.02	n.d.	n.d.	0.03*	0.02*	0.06 ± 0.03	
0.06 ± 0.02	n.d.	0.01 ± 0.004	0.01*	0.01*	0.07 ± 0.04	
1.74 ± 0.71	0.02*	0.4 ± 0.13	0.28 ± 0.18	0.27 ± 0.11	1.82 ± 0.59	
0.03*	n.d.	0.01*	0.02*	0.01*	0.04 ± 0.02	
0.06 ± 0.02	n.d.	0.04 ± 0.01	0.03*	0.04 ± 0.01	0.09 ± 0.07	
0.12 ± 0.11	n.d.	0.07 ± 0.01	0.02 ± 0.01	0.04 ± 0.04	0.33 ± 0.32	
0.21 ± 0.08	n.d.	0.12 ± 0.02	0.07 ± 0.03	0.09 ± 0.03	0.46 ± 0.24	
0.03*	n.d.	n.d.	n.d.	n.d.	n.d.	
0.04 ± 0.01	n.d.	0.02 ± 0.004	0.03*	0.02*	0.05 ± 0.03	
0.04*	n.d.	n.d.	n.d.	0.01*	0.04 ± 0.01	
0.05 ± 0.02	n.d.	n.d.	0.02*	0.02*	0.06 ± 0.03	
0.16 ± 0.02	n.d.	0.02 ± 0.01	0.05 ± 0.004	0.05 ± 0.01	0.15 ± 0.03	
0.37 ± 0.06	n.d.	0.14 ± 0.03	0.12 ± 0.03	0.14 ± 0.03	0.61 ± 0.09	
2.34 ± 0.44	0.05 ± 0.01	0.59 ± 0.12	0.67 ± 0.13	0.63 ± 0.08	3.56 ± 0.4	
Notes.

FA fatty acids

SFA saturated fatty acids

MUFA monounsaturated fatty acids

PUFA polyunsaturated fatty acids

n.d. not detected

Data with asterisks indicate that only one data value was available, so it was not possible to calculate the standard deviation. The bold values indicate the sum of the different fatty acid types.

x Species not considered in the statistical analysis (n < 3).

Regarding the total concentration of fatty acids, the highest average values were recorded in X. gladius (3.56 ± 0.4 mg 20 mg−1 DW), R. pretiosus (2.34 ± 0.44 mg 20 mg−1 DW) and L. flavobrunneum (3.38 ± 0.57 mg 20 mg−1 DW), intermediate mean values in K. audax (1.67 ± 0.15 mg 20 mg−1 DW), G. melampus (1.1 ± 0.21 mg 20 mg−1 DW), T. albacares (0.67 ± 0.13 mg 20 mg−1 DW), T. alalunga (0. 59 ±  0.12 mg 20 mg−1 DW) and T. obesus (0.63 ± 0.08 mg 20 mg−1 DW), and the lowest values in P. glauca (0.13 ± 0.01 mg 20 mg−1 DW), I. oxyrinchus (0.10 ±  0.01 mg 20 mg−1 DW), S. zygaena (0.05 ± 0.01 mg 20 mg−1 DW), and C. hippurus (0.06 ± 0.004 mg 20 mg−1 DW). For more details regarding the fatty acid contents determined for each species, see Table 2.

When considering the saturated fatty acids, palmitic acid (C16:0) was observed in the greatest amounts in X. gladius (0.59 ± 0.4 mg 20 mg−1 DW), followed by T. alalunga (0.30 ± 0.24 mg 20 mg−1 DW). The monounsaturated fatty acid found in the highest concentrations was oleic acid (C18:1n9) in species of L. flavobrunneum (2.04 ± 0.88 mg 20 mg−1 DW), R. pretiosus (1.36 ±  1.04 mg 20 mg−1 DW) and X. gladius (1.31 ± 0.84 mg 20 mg−1 DW). For the polyunsaturated fatty acids, the eicosapentanoic (EPA, C20:5n3) and docohexaenoic acids (DHA, C22:6n3) were abundant in X. gladius (0.09 ± 0.07 mg 20 mg−1 DW) and L. flavobrunneum (0.08 ± 0.04 mg 20 mg−1 DW) (Table 2).

The PCoA showed a high spatial variability in the fatty acid profiles of the diverse highly migratory species analyzed, explaining almost 77% of the variance between both axes. PCO1 explained 66% of the variation in the fatty acid profile, while the PCO2 axis explained only 10.8% of the variation (Fig. 3). On the PCO1 axis, a clear difference in the fatty acid profiles between species was observed (e.g., L. favobrunneum, X. gladius, R. pretiosus vs. L. nasus, L. imperialis). In addition, there are species that showed a high similarity in their fatty acid profiles, such as C. hippurus and I. oxyrinchus. On the other hand, species of the Thunnus genus presented a high variability, though some were characterized by a degree of similarity. In turn, despite the high number of samples analyzed of X. gladius, these specimens did not represent a wide variability between the PCO axes.

Figure 3 Principal coordinates analysis (PCoA) of muscle tissue of N = 13 highly migratory species off the coast of Chile.

Different colors and forms represent the different species.

The MDS revealed differences in the distribution of the fatty acid profiles of the analyzed species (two dimensions with stresses of 0.1; Fig. S1). In particular, notorious groupings were observed for each species, in which their fatty acid profiles were conspicuously separated and/or distant depending on the species: (i) X. gladius: green triangles; (ii) R. pretiosus: yellow border triangles; and (iii) L. flavobrunneum: blue inverted triangles. They were also clearly grouped according to taxa: (i) X. gladius; (ii) R. pretiosus; (iii) L. guttatus; (iv) T. alalunga, T. albacares, T. obesus, L. flavobrunneum, G. melampus; (v) L. nasus, A. vulpinus, P. glauca, I. oxyrinchus, S. zygaena; and finally, very close to these species of chondrichthyans, C. hippurus. For further details see Fig. S1.

The similarity analysis (ANOSIM) confirmed that the fatty acid profiles of the evaluated species were statistically significant, and very different among them, with a global coefficient of determination (Global R) that includes all the paired comparisons of R = 0.761 (values of R close to 1 indicate very different fatty acid profiles among species; with a significance level of 0.1%; and 999 permutations). For details regarding all of the paired comparisons, see Table S1.

In turn, the similarity percentage analysis (SIMPER) indicated that the greatest contribution to this differentiation was provided by the fatty acids oleic (C18:1n9), palmitic (C16:0), stearic (C18:0), eicosenoic (C20:1), and DHA (C22:6n3). The SIMPER also demonstrated that the species X. gladius, R. pretiosus, L. flavobrunneum, I. oxyrinchus and C. hippurus contributed the most to the significant differences found among the fatty acid profiles. For more details see Table 3.

Table 3 Analysis of similarity percentage (SIMPER) of the fatty acid profiles of muscle tissue of N = 13 highly migratory species off the coast of Chile.

Specie	Similarity percentage (%)	Fatty acid	Av.
Abund.	Av.
Sim.	Sim/SD	Contrib. %	Cum. %	
Alopias vulpinus	87.06	Palmitic (C16:0)	7.27	45.30	10.14	52.04	52.04	
		Stearic (C18:0)	6.12	37.76	9.62	43.38	100.00	
Coryphaena hippurus	93.26	Palmitic (C16:0)	5.52	27.35	14.13	29.33	29.33	
		Oleic (C18:1n9)	5.59	27.18	13.20	29.14	58.47	
		Stearic (C18:0)	4.59	22.89	15.66	24.54	83.01	
Isurus oxyrinchus	84.19	Palmitic (C16:0)	5.82	27.95	8.85	33.20	33.20	
		Stearic (C18:0)	4.45	21.24	8.30	25.22	58.42	
		Oleic (C18:1n9)	4.47	17.99	1.68	21.37	79.79	
Lamna nasus	87.98	Palmitic (C16:0)	7.39	43.69	10.21	49.66	49.66	
		Stearic (C18:0)	4.86	28.25	9.12	32.11	81.77	
		Oleic (C18:1n9)	3.98	16.04	1.05	18.23	100.00	
Lampris guttatus	73.62	Oleic (C18:1n9)	5.31	17.72	8.03	24.07	24.07	
		Palmitic (C16:0)	4.36	14.71	5.61	19.99	44.06	
		Miristic (C14:0)	3.48	10.46	2.40	14.21	58.28	
Lepidocybium flavobrunneum	78.76	Oleic (C18:1n9)	7.98	28.26	9.94	35.89	35.89	
		Eicosenoic (C20:1)	2.48	8.41	6.42	10.68	46.57	
		DHA (C22:6n3)	2.38	7.35	4.10	9.33	55.90	
Luvarus imperialis	92.46	Oleic (C18:1n9)	5.67	29.26	24.98	31.65	31.65	
		Palmitic (C16:0)	5.71	28.65	20.30	30.99	62.64	
		Stearic (C18:0)	4.89	24.90	25.42	26.93	89.56	
Prionace glauca	69.35	Palmitic (C16:0)	6.15	28.96	2.83	41.76	41.76	
		Estearic (C18:0)	5.11	24.00	2.79	34.60	76.36	
		Oleic (C18:1n9)	3.06	7.46	0.73	10.76	87.13	
Ruvettus pretiosus	77.02	Oleic (C18:1n9)	7.23	22.75	4.31	29.54	29.54	
		Eicosenoic (C20:1)	2.78	8.66	3.68	11.25	40.79	
		DHA (C22:6n3)	2.29	7.08	7.61	9.19	49.98	
Thunnus alalunga	71.54	Palmitic (C16:0)	4.94	13.99	3.89	19.55	19.55	
		Oleic (C18:1n9)	4.16	11.48	3.59	16.04	35.60	
		Eicosenoic (C20:1)	3.25	8.59	3.29	12.01	47.60	
Thunnus albacares	72.55	Palmitic (C16:0)	5.29	20.10	8.58	27.70	27.70	
		Oleic (C18:1n9)	5.20	18.86	7.57	25.99	53.70	
		Stearic (C18:0)	3.72	13.55	4.66	18.67	72.37	
Thunnus obesus	76.75	Oleic (C18:1n9)	5.52	17.21	4.67	22.42	22.42	
		Palmitic (C16:0)	4.79	15.94	4.79	20.77	43.19	
		Stearic (C18:0)	3.30	11.01	6.69	14.34	57.53	
Xiphias gladius	86.1	Oleic (C18:1n9)	6.22	17.97	5.49	20.88	20.88	
		Palmitic (C16:0)	4.22	12.48	8.74	14.5	35.37	
		Stearic (C18:0)	2.59	7.56	8.27	8.78	44.15	
Notes.

Av. Abund average abundance of each fatty acid

Av. Sim average similarity contributed by the fatty acid

Sim/SD similarity ratio to standard deviation

Contrib% percentage of contribution of the fatty acid to the overall similarity

Cum.% additive overall similarity percentage

Discussion

In the present study, we effectively determined the fatty acid compositions of the highly migratory species obtained from the fishing fleets off the coast of Chile. The physiological attributes of the fatty acid biomolecules of marine fishes in their environment present a degree of specificity according to the feeding or trophic habit of each species, and can therefore reflect the environmental conditions experienced by the organisms (due to the type of food consumed) under a spatial scale of geographic regions in the Southeastern Pacific Ocean (SEPO) (Gong et al., 2018; Lazo-Andrade et al., 2021; Quispe-Machaca et al., 2021). In this context, our findings are the first step necessary to determine the traceability of highly migratory fishery resource species from the SEPO off the coast of Chile through fatty acid profile analyses. This method could thus be used to distinguish among species and/or origin, based on the fatty acid profiles of their tissues (i.e., muscle and/or fillet), which in this case constitute the main part of the body intended as food for human consumption (Barría et al., 2020; Leal et al., 2015; Ricardo et al., 2015). The traceability of fishery resources through the analysis of fatty acids, as a potential tool to determine the geographical origin of the species, has important implications for their sustainable exploitation within an ecosystem and precautionary approach (Guzmán-Rivas et al., 2021; Lazo-Andrade et al., 2021). Together with DNA analyses, this biochemical tool (e.g., fatty acids) could allow us to quickly and efficiently determine and/or distinguish the species and its geographical origin (capture zone) immediately after the unloading process at the port and subsequently at the processing plant (Leal et al., 2015; Santos et al., 2023; Stowasser, DW & Collins, 2009). Additionally, identifying the nutritional profile of fishery resources allows us to promote the exploitation of certain species that have a healthier and more sustainable origin throughout the entire production chain (i.e., from capture at sea, landing, processing and finally making it to the consumer’s plate) (Gong et al., 2018; Molkentin et al., 2015).

In the case of highly migratory fishery resource species, our study highlights the high variety of fatty acids (saturated; mono-unsaturated; poly-unsaturated) in swordfish (X. gladius), oilfish (R. pretiosus), tuna (albacore: T. alalunga; yellowfin: T. albacares; big eye: T. obesus) and fish (escolar: L. flavobrunneum; butterfly kingfish: G. melampus). On the contrary, a low variety of fatty acids was found in the evaluated species of sharks and/or cartilaginous fish (porbeagle: L. nasus; common thresher: A. vulpinus; blue: P. glauca; shortfin mako shark: I. oxyrinchus; hammerhead: S. zygaena), in which only saturated fatty acids predominated. These differences in the fatty acid profiles of the evaluated species can be explained not only by differences associated with a phylogenetic and/or taxon trait (i.e., comparisons between bony fishes vs. cartilaginous fishes and/or sharks), but also by the geographic origin of the species (i.e., high latitude species vs. low latitude species) (Pethybridge et al., 2010; Prato & Biandolino, 2012; Zhang et al., 2023). In turn, the low variety of fatty acids in the shark species analyzed in this study may be related to an ancestral evolutionary trait of common diversification in tropical environments, where saturated fatty acids have predominated (Shadwick, Farrell & Brauner, 2015). These fatty acids in marine organisms are highly important in cellular homeostasis processes and physiological processes that adapt to different temperatures.

From an ecophysiological point of view, particularly regarding the structure, function and interaction with the environment, fatty acids play fundamental roles in the biophysical process of homeoviscous adaptation in fishes (Ernst, Ejsing & Antonny, 2016). This complex process allows fishes to maintain an optimal level of fluidity in their cell membrane and body tissue, depending on the water temperature (i.e., in cold temperatures, polyunsaturated fatty acids predominate vs. warm temperatures where saturated fatty acids predominate; Chu et al., 2024; Howell & Matthews, 1991). This may explain the high displacement capacity of these species from latitudes of warm to cold temperate waters of the Humboldt Current System due to the high homeoviscous capacity present in their tissues and membranes (Shadwick, Farrell & Brauner, 2015; Malekar et al., 2018; Leigh, Papastamatiou & German, 2017; Lazo-Andrade et al., 2023). Also, similar biochemical adaptations linked to migratory behavior have been described for species of sharks and swordfish (Nogrady, 2023; Lazo-Andrade, Barría & Urzúa, 2024), which move from shallow habitats to the deep sea, with cold waters, in search of prey rich in fats (Royer et al., 2023).

In addition, within an ecosystem approach, differences or similarities in the fatty acid profiles can be associated with the degree of trophic specialization of each organism (e.g., swordfish are considered specialists, while sharks are considered generalists; Lazo-Andrade et al., 2021; Munroe, Simpfendorfer & Heupel, 2014), their trophic spatial position (e.g., species occupying the same trophic level) (Kainz et al., 2017), as well as the type of food they consume (e.g., large or small prey) within the marine food web (Markaida & Sosa-Nishizaki, 2010; Loor-Andrade et al., 2017). Our findings based on the similar fatty acid profiles of some species (e.g., I. oxyrinchus, C. hippurus, P. glauca) could indicate that these species may present a similar trophic level (Rosas-Luis et al., 2017; Paez-Rosas et al., 2018), and also a voracious feeding behavior (Letelier et al., 2009; Lopez, Meléndez & Barría, 2009; Lopez, Barría & Meléndez, 2012) characterized by the consumption of large prey rich in saturated fatty acids (Rosas-Luis et al., 2017; Barría et al., 2024). In this sense of feeding ecology, cephalopod species has been described as the main prey item of billfish and swordfish species, which is consequently reflected in the fatty acid profiles of their tissues and/or organs (Quispe-Machaca et al., 2021; Quispe-Machaca et al., 2022; Hu et al., 2022; Guzmán-Rivas et al., 2023). In particular, the swordfish X. gladius has demonstrated a specialist hunting habit on the jumbo squid Dosidicus gigas (Ibáñez, González & Cubillos, 2004), first stabbing and then cutting their prey with the sword of their mouth structure (Berkovitz & Shellis, 2017; Preti et al., 2023). Recent findings utilizing fatty acids as biomarkers of the trophic interaction between these two highly migratory resources of the southern Humboldt System revealed the degree of preference swordfish have for preying on jumbo squid (D. gigas), mainly their digestive gland, which is an organ rich in lipids and fatty acids (Hu et al., 2022; Lazo-Andrade et al., 2023; Guzmán-Rivas et al., 2023).

Future comparative studies should consider that the fatty acid profiles of highly migratory species may be influenced, not only by their feeding habits and degrees of specialization in terms of the capture and consumption of fatty prey present in this geographic marine area (for the concept of trophic migration of highly migratory species in the Humboldt System, see: Hu et al., 2022; Lazo-Andrade, Barría & Urzúa, 2024; Massing et al., 2022), but also by the environmental factors and prevailing oceanographic conditions (e.g., water temperature, upwelling events, oxygen levels, food availability) of the study site. These environmental conditions may vary temporally (between seasons and years), potentially increasing the variability in the profile of these marine top predators (Cretton et al., 2023). Finally, these data could be used as reference information in other fisheries across Chile, as well as to characterize the nutritional condition or trace geographic origin. Consequently, future studies with an integral and ecosystemic approach should incorporate: (i) other organs of the body, such as the liver and the gonad, which are also considered potential fishery products with high added value; (ii) the capture area and sex of the organisms; (iii) a complementary tool for molecular identification through the genetic analysis of species, (iv) a longer time scale for the study.

Conclusion

The present study revealed notorious differences in the fatty acid compositions of the muscle of HMRS of the SEPO, off the coast of Chile. We corroborated our hypothesis that the fatty acid profiles may vary, not only depending on the prey consumed through a characteristic hunting habit and/or behavior, but also due to their phylogeny and/or taxonomic group of evolutionary origin. Swordfish also had the highest abundance and concentration of fatty acids, followed by escolar fish and oilfish, while the hammerhead shark had the lowest abundance and concentration of fatty acids. These results are the first step needed to characterize the fatty acid profiles of the diverse fisheries in Chile. Within an ecosystem approach, our findings help us to understand how essential biomolecules of fatty acids are transferred through the marine food web in the SEPO, revealing the diet type and/or feeding habits of HMRS considered as marine top predators. Furthermore, identifying the fatty acid profiles of fishery resources at a spatial level could be relevant data for their management and conservation, particularly for those resources that are overexploited and also have a critical nutritional importance for human consumption.

Supplemental Information

Supplemental Information 1 Multidimensional scaling analysis (MDS) of muscle tissue of N = 13 highly migratory species off the coast of Chile

Different colors and forms represent the different species.

Supplemental Information 2 The raw measurements of fatty acids

Supplemental Information 3 Analysis of similarities (ANOSIM) of the muscle tissue of N = 18 highly migratory species off the coast of Chile. Global statistical test (R = 0.761)

*Comparisons: the following species were not included in the analysis, Makaira indica, Tetrapturus audax, Makaira indica, Sphyrna zygaena due to a lack of replication (only one specimen was analyzed).

We sincerely thank the editor and reviewers for their constructive criticism and valuable suggestions. Special thanks to Christine Harrower for correcting the English and improving this manuscript. We also thank the scientific observers for their assistance aboard fishing vessels.

Additional Information and Declarations

Competing Interests

Author Contributions

Animal Ethics

Data Availability

The authors declare there are no competing interests.

Fabián Guzmán-Rivas conceived and designed the experiments, performed the experiments, analyzed the data, prepared figures and/or tables, authored or reviewed drafts of the article, and approved the final draft.

Marco Quispe-Machaca conceived and designed the experiments, performed the experiments, analyzed the data, prepared figures and/or tables, authored or reviewed drafts of the article, and approved the final draft.

Jorge Lazo conceived and designed the experiments, performed the experiments, analyzed the data, prepared figures and/or tables, authored or reviewed drafts of the article, and approved the final draft.

Juan Carlos Ortega conceived and designed the experiments, performed the experiments, analyzed the data, prepared figures and/or tables, authored or reviewed drafts of the article, and approved the final draft.

Sergio Mora conceived and designed the experiments, analyzed the data, authored or reviewed drafts of the article, and approved the final draft.

Patricio Barría Martínez conceived and designed the experiments, analyzed the data, authored or reviewed drafts of the article, and approved the final draft.

Ángel Urzúa conceived and designed the experiments, performed the experiments, analyzed the data, prepared figures and/or tables, authored or reviewed drafts of the article, and approved the final draft.

The following information was supplied relating to ethical approvals (i.e.,  approving body and any reference numbers):

The Vice Rectorate for Research and Postgraduate Studies of the Universidad Católica de la Santísima Concepción (UCSC) and the Undersecretariat for Fisheries and Aquaculture (SUBPESCA) of Chile granted full approval for this research (Law 20.380/ VRIP/UCSC/ 27/ 2023)

The following information was supplied regarding data availability:

The raw data are available in the Supplementary Files.

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
