# Peer review of "Fatty acid profiles of highly migratory resources from the Southeastern Pacific Ocean, Chile: a potential tool for biochemical and nutritional traceability"

_PeerJ, doi:10.7717/peerj.19101_

## Round 0.1 · original submission · Major Revisions

· Academic Editor

Major Revisions

The work is interesting but should be thoroughly revised following the indications of the reviewers. Special attention should be paid to the fact that some results are presented for a very low number of specimens, which does not allow any conclusion to be drawn. Perhaps, these data should be omitted.

·

Basic reporting

line 46: change to "oilfish (Ruvettus pretiosus) and escolar fish (Lepidocybium flavobrunneum)"

line 53: is this meant to be "notable" here?

line 315: please clarify the last part of this sentence.

line 354: delete the repeated last word in the sentence.

Figure 3 and 4: On both graphs please change the title of the legend into English.

Table 1 title: change "specie" to "species".

Experimental design

line 181-183: can you explain some more about the differences in where the muscle samples were taken, as I would think this would make a big difference in the fatty acids and might be a confounding factor (is the method used in this study the regular method used in past studies? Is it not possible to choose a kind of muscle that can be taken from the same place on the cartilaginous fish and the bony fish?).

line 228-229: according to Table 1, five of the species were represented by only one or two individuals (n=1-2). I would say these should not have been used in a statistical analysis like PERMANOVA as the sample size is not high enough. Swordfish had by far the greatest sample size (n=27), so maybe the high diversity of fatty acids found for this species (line 233-234) just reflected the diversity of different individuals tested? Especially in a case like this, it should have been strived to have an equal sample size. For the species with n=1, the results can certainly only apply to either males or females (not both), again highlighting the limitation of using such a limited sample size in some of the cases here.

Validity of the findings

line 294-296: statements like "These findings confirm the importance of using this method for the biochemical traceability of species and fishery products of highly migratory resources in Chile" are not valid when the analyses had some species with only n=1 (i.e. it is not valid for the result from a single fish to be incorporated into a generalisation up to the level of the national fishery). Ideally, the analysis should have waited (specimens frozen) until a decent sample size of all species had been collected, then the same number of individuals from each species used (e.g. n=7 or 8 or something like that).

If some species with low abundances were taken out of this study, and some samples of the more abundant species randomly removed from the analysis, then the analysis could be redone here with an equal sample size. If this were done the findings would not be so broad (e.g. several of the species would no longer be incorporated at all), but in terms of the statistics, it would be one way to made the dataset's analysis valid.

Additional comments

line 41: please could you clarify here whether the "origin" is referring to the species of fish or the region where it was fished.

line 55: what is meant by "best"? Does this mean the most healthy for human consumption, or something like that?

line 109: why do the fatty acids in species low in the food chain make it possible to trace the geographic origin of the fish?

line 212: which of the statistical tests were done with Rstudio?

line 263: normally either an MDS ordination is done, or a PCO ordination, not both. The same grouping can be seen in Fig. 3 and Fig. 4, so I am not sure how much more information is added by including the MDS after already showing the PCO. I would delete Fig. 4 and discuss about the groupings of the different points from Fig. 3.

line 297: please explain some more here about how the method can be used to distinguish between different geographic origins.

line 357-358: does the digestive gland have a very diverse set of fatty acids? (as this study found the swordfish muscle had the most diverse fatty acids).

Figure 3: please mention in the methods text that the data were square root transformed prior to analysis (and confirm whether that transformation was done for the data in all the other analyses).

·

Basic reporting

Even if I’m not a native English speaker, I feel the present version of the manuscript suffers from some language issues, and it may benefit from a correction by a fluent speaker.
Literature references provide sufficient field background/context, however is composed mainly (65.5%) by old information (more than five years). Some of them are important, such as literature regarding statistical analysis and some classic themes, but authors should spend some effort on recent literature from the last five years.
Structure, figures and table are relevant to the article, however need improvement. Figure 1 lacks images of bony and cartilaginous fish species with good resolution. The authors should improve it or think about other options. Alternatively, they can use scientific drawings of each species instead of low-resolution images. Figures 2, 3 and 4 are good.
Regarding Table 1, what is the logic of the order of species? Authors should present species in descending order in the number of individuals.
Regarding Table 2, authors must indicate what bold values means; authors should present species with its FAs concentrations in the same order as Table 1; the orientation of table 2 must be landscape; and authors must indicate that "Average values and standard deviation" are both by each fatty acid and by types of fatty acids (SFA, MUFA, PUFA-n3, PUFA-n6, PUFA).

Experimental design

No comment

Validity of the findings

The link between the raised hypothesis (L148 - L153) with the conclusions (L369 - L374) must be improved. As it is presented is disconnected. Thus, authors must rewrite in order to emphasize in the text if their hypothesis were accepted or refuted.

Reviewer 3 ·

Basic reporting

No comment

Experimental design

No comment

Validity of the findings

No comment

Additional comments

The manuscript is concise, clear and well-written. Below are my suggestions for improving the manuscript:

Title: The term "biochemical traceability" should be defined in the document. Perhaps a more appropriate term is "biochemical determination", since at first glance "biochemical traceability" may be understood as the possibility of tracing the biochemical processes through which foods have passed. In addition, if "nutritional traceability" is used as a synonym for "food traceability", this should be stated. Otherwise, the definition should appear in the introduction.

Abstract: it is necessary to state the objective. Likewise, it is necessary to briefly include information on the methodology; for example, the method used for the determination of the fatty acid profiles. The last sentence of the abstract synthesizes the results well; however, it does not address the applications or theoretical implications of the results (conclusion).

Methods: in the 2013 line, the authors could be more specific by clarifying what they mean by “advanced statistical analyses”.

Results: in lines 236 to 246, the species Katsuwonus pelamis and Gasterochisma melampus are not mentioned, in which categories are they classified (intermediate, low diversity)? The data for total fatty acid concentration (line 248) are expressed in mg, while those for saturated fatty acids are expressed in 20 mg-1 DW, is there any reason why the same unit is not used? In the “MDS and PCoA” section, Figure 3 (PCoA) is not described; the results that appear in that figure must be described.

Discussion: if the species studied are highly migratory, wouldn't it be expected that they would feed on organisms from different areas and, therefore, it would not be possible to identify their geographical origin based on their fatty acid profile? The authors are requested to discuss this. Also, it is mentioned that the differences in the fatty acid profiles of the evaluated species can be explained from a phylogenetic and geographic point of view; what about the environment, are there articles that mention whether changes in environmental conditions can modify the fatty acid profile?

Conclusions: the last paragraph of the discussion could be integrated into the conclusion and perhaps be complemented by mentioning what are the challenges of characterizing the fatty acid profile of the different species along the Chilean coast, which species need priority characterization, etc.

Other comments:

In line 94 a quotation is needed to support the information contained in that sentence.
In line 96 the authors could briefly define what the “unload process” consists of.
Line 129: the full species name Xiphias gladius has already been written in full in line 86, so from line 86 onwards the author could write X. gladius; check this throughout the document for the different species (line 233, 237, 238, etc.)... See: https://www.aje.com/arc/editing-tip-scientific-names-species/: “When the same name is used more than once in a paper, the first letter of the genus (still capitalized) may be used as an abbreviation in the second and subsequent uses of the name, but the rest of the name is not abbreviated (R. marianna, D. involucrata). In particular, the name is commonly written out in full when it first appears in the abstract and then abbreviated in the rest of the abstract using the convention shown above. The name is again written out in full when it first appears in a subsequent section of the paper (typically, the introduction) and is then abbreviated upon further use.”
The wording of the information in lines 147-152 is not entirely clear, points "ii" and "iii" are hypotheses that will be tested in the study?
In line 185, -80 ºC is written with a space between the number and the unit, while in line 188 it is not. It is necessary to choose a format and homogenize it in the document.
Line 232: the abbreviation to be used for dry weight (DW) has to be placed in line 193. Thus, in line 232 the authors could use DW instead of dry weight.
Line 261: the authors cite Figure 3, is the citation of Figure 3 necessary?
Line 264: the meaning of MDS is already defined in line 216, it is not necessary to write it in full.
Line 266: It is not clear what is meant in this sentence "In particular, notorious groupings were observed for each species, in which their fatty acid profiles were conspicuously separated and/or distant depending on the species: i) swordfish (X. gladius: green triangles), ii) oilfish (R. pretiosus: yellow border triangles), and iii) escolar fish (L. flavobrunneum: blue inverted triangles)." . You mean the biological replicates of X. gladius, R. pretiosus and L. flavobrunneum species were grouped by species or in a single group?
Line 270: it is not necessary to include both the scientific name and the common name, one of them is enough; check throughout the document.
If the illustrations used in Figure 1 were not designed by the authors or are not copyright free, it is important to indicate the source from which they were obtained. In addition, the illustrations of Alopias vulpinus and Sphyrna zygaena have a very low quality.
In Figure two, it should read "W" instead of "O" (west instead of "oeste"). In the figure legend, it is worth mentioning that different colors and shapes represent different species.
Figures 3 and 4 should read "species" instead of "especie". Information regarding transformation and resemblance are not necessary in Figures 3 and 4, they could be included in the methods section and/or figure captions.
In the title of table 1 it should read “species” and not “specie”.

---

## Round 0.2 · Minor Revisions

· Academic Editor

Minor Revisions

Thank you very much for accepting the reviewers' suggestions and incorporating them into the manuscript. However, one reviewer suggested minor modifications that could improve the standard of the manuscript. Please review the attached file.

·

Basic reporting

No comment

Experimental design

No comment

Validity of the findings

No comment

Additional comments

The authors addressed all corrections and suggestions made by the reviewers. Therefore, I recommend publishing the manuscript.

General comments:

L238: Change "Individuals" to "Species".
L239: Change "speciemens" to "species".

Reviewer 3 ·

Basic reporting

No comment, the authors have considered the observations and suggestions made in the first version of the manuscript.

Experimental design

No comment, the authors have taken into account the observations and suggestions made in the first version of the manuscript.

Validity of the findings

As a last suggestion, I recommend mentioning some additional examples (based on your results) of species that show fatty acid profiles more similar to each other because they occupy the same trophic level or examples of groups of species that have similar fatty acid profiles because of the type of food they consume (e.g., large or small prey). More examples would reinforce the idea that there is an association between the fatty acid profile and each species' feeding or trophic habit.

Additional comments

Some minor suggestions were added to the clean version of the manuscript (see attached document).

Annotated reviews are not available for download in order to protect the identity of reviewers who chose to remain anonymous.

---

## Round 0.3 · accepted · Accept

· Academic Editor

Accept

The authors have satisfactorily addressed all the reviewers' comments. I confirm that I have assessed the revision myself and find the current version to meet the required standards. Therefore, this manuscript is ready for publication.

Reviewer 3 ·

Basic reporting

No comment

Experimental design

No comment

Validity of the findings

No comment

Additional comments

The authors have addressed the corrections and recommendations provided by the reviewers. Thus, I suggest proceeding with the publication of the manuscript.